# Modeling and Integrated Optimization of Power Split and Exhaust Thermal Management on Diesel Hybrid Electric Vehicles

**Jinghua Zhao [1,2], Yunfeng Hu [1], Fangxi Xie [1], Xiaoping Li [1], Yao Sun [1], Hongyu Sun [2] and Xun Gong [1,3,*]**

[1] State Key Laboratory of Automotive Simulation and Control, NanLing Campus, Jilin University, Changchun 130025, China; zhaojh08@mails.jlu.edu.cn (J.Z.); huyf@jlu.edu.cn (Y.H.); xiefx2011@jlu.edu.cn (F.X.); lixp2008@jlu.edu.cn (X.L.); syao@jlu.edu.cn (Y.S.)

[2] Computer College, Jilin Normal University, Siping 136000, China; hongyu@jlnu.edu.cn

[3] School of Artificial Intelligence, Jilin University, Changchun 130025, China

[*] Correspondence: gongxun@jlu.edu.cn

**Abstract:** To simultaneously achieve high fuel efficiency and low emissions in a diesel hybrid electric vehicle (DHEV), it is necessary to optimize not only power split but also exhaust thermal management for emission aftertreatment systems. However, how to coordinate the power split and the exhaust thermal management to balance fuel economy improvement and emissions reduction remains a formidable challenge. In this paper, a hierarchical model predictive control (MPC) framework is proposed to coordinate the power split and the exhaust thermal management. The method consists of two parts: a fuel and thermal optimized controller (FTOC) combining the rule-based and the optimization-based methods for power split simultaneously considering fuel consumption and exhaust temperature, and a fuel post-injection thermal controller (FPTC) for exhaust thermal management with a separate fuel injection system added to the exhaust pipe. Additionally, preview information about the road grade is introduced to improve the power split by a fuel and thermal on slope forecast optimized controller (FTSFOC). Simulation results show that the hierarchical method (FTOC + FPTC) can reach the optimal exhaust temperature nearly 40 s earlier, and its total fuel consumption is also reduced by 8.9%, as compared to the sequential method under a world light test cycle (WLTC) driving cycle. Moreover, the total fuel consumption of the FTSFOC is reduced by 5.2%, as compared to the fuel and thermal on sensor-information optimized controller (FTSOC) working with real-time road grade information.

**Keywords:** diesel hybrid electric vehicle (DHEV); power split; exhaust thermal management; nonlinear model predictive control (NMPC)

## 1. Introduction

Diesel vehicles have a higher performance in fuel economy and reliability as compared to gasoline vehicles. However, their high $NO_x$ and particulate matter (PM) emissions remain a concern [1]. The world light test cycle (WLTC) [2] was implemented in the European Union (EU) as of September 2017. Except for the stricter pollutants limit, one of the most significant updates for the China 6/Euro 6 is that NEDC is replaced by WLTC. Compared to NEDC, the fuel consumption and energy demand for WLTC is 1–11% and 26–44% higher, respectively. Over the last decade, researchers have continuously focused optimization efforts on well-defined but unrealistic driving cycles, such as WLTC, to reduce the pollutant emissions of diesel vehicles. In fact, real driving emission tests recently implemented, such as the portable emission measurement system (PEMS), which detect the emissions emitted during every-day driving, can far exceed the legislative limits [3]. Researchers are working hard for more advanced technologies on fuel economy improvement and emissions reduction. However, it will be difficult to achieve substantial

improvements while relying only on the particular technology of diesel vehicles themselves. Hybrid powertrains can provide an additional degree of freedom for these improvements on fuel economy and emissions [4]. Recently, an authoritative magazine highlighted that the advantages of pure electric vehicles on fuel economy and emissions from their production and use are not obvious, because the electricity supply in most countries still primarily comes from burning coal and natural gas. For the foreseeable future, the power source of vehicles will be characterized by a mix of solutions involving internal combustion engines, batteries and hybrid powertrains, of which hybrid powertrains have the greatest development potential [5,6]. The working mode of diesel hybrid electric vehicles (DHEVs) can not only improve fuel economy but also reduce emissions, including that of $NO_x$, PM and $CO_2$, etc. There have been many studies on HEVs, but the establishment of and discussion around complete models, including aftertreatment systems, for DHEVs are lacking.

The key to controlling hybrid electric vehicles (HEVs), including DHEVs, is the power split between the engine and the motor for coordinating fuel consumption and emissions [7]. Strategies for the power split are generally classified into two categories: rule-based methods and optimization-based methods. Rule-based methods are empirically tuned through offline data analysis to attain a better performance over a specified driving cycle. In [8], a Sugeno–Takagi fuzzy logic controller for HEVs with a parallel configuration is developed, which employs a set of rules, such as driver command, the battery state of charge (*SoC*) and motor speed, to improve the overall efficiency [9]. In [10], a neural network controller for Toyota Prius using offline training is developed, and its experimental results show that this controller can achieve less variance of *SoC* and better fuel economy. Rule-based methods have some obvious advantages, including easy implementation, higher computational efficiency, lower hardware cost, etc. Nevertheless, their inherent calibration characteristics cannot adapt to different driving scenarios, and their poor anti-interference also limits real-world performance and application. Optimization-based methods primarily solve the power split by minimizing cost functions of optimization problems. The cost functions are generally defined by a set of performance indices, including fuel consumption, emissions, *SoC*, etc. The solutions to optimization problems, i.e., control actions, can be obtained through analysis or numerical calculation. Dynamic programming (DP) is a noncausal solution for optimization problems, which requires a detailed understanding of the driving cycle information. Therefore, it has a heavy computational burden and does not have real-time performance, but its optimization results can be utilized as benchmarks for other control methods. In [11], a DP method globally optimizing the cost function is utilized to derive a gear-shift and power split strategy. In [12,13], an equivalent consumption minimization strategy (ECMS) for real-time implementation is proposed, which uses an equivalent factor constructing a cost function to combine the electricity and the fuel consumption of the HEVs. In [14], an improved F-ECMS strategy using variable geometry turbocharger (VGT) and exhaust gas recirculation (EGR) to control fuel economy for DHEVs is proposed. Model Predictive Control (MPC) is another commonly used optimization method, which can utilize preview information for optimization control in a rolling horizon. In [15], a power split strategy for HEVs based on MPC is proposed. Sequential Quadratic Programming (SQP) [16] can effectively solve optimization problems under the MPC framework. However, its solution process for multi-objective problems with constraints requires a heavy computational burden. How to combine the rule-based and optimization-based methods to solve the complex problem of the power split with less computational burden remains a difficult problem.

Compared to ordinary gasoline HEVs, control strategies for DHEVs have to take into account not only the fuel economy at the power split level but also the original emissions of engines [17]. In [18], a control strategy based on a constant weighting factor for $NO_x$ emission is verified by means of a hardware-in-the-loop (HIL) platform of DHEVs. Some studies on the transient energy and emission control of DHEVs are also discussed. They focus on how to use motors instead of engines under transient conditions so as to better

coordinate fuel and emission performance [19,20]. In [20], a constant weighting factor related to $NO_x$ and PM emissions is designed, and a power split strategy based on the DP method is proposed. In [6], an ECMS method for the power split based on the online adaptive weighting factors for the objective pollutant is proposed under the PEMS test. This method can minimize fuel consumption while tracking a specific $NO_x$ emission level and sustaining the battery *SoC*. In [21], a control method utilizing the fuel injection mass, the fuel injection timing and the power split rate for improving the fuel economy and the $NO_x$ emission of DHEVs is proposed. In [22], an MPC method controlling a diesel-electric marine to suppress the transient fuel consumption and emissions of diesel engines is proposed. Compared with the three-way converter (TWC) of gasoline vehicles, the aftertreatment system of diesel vehicles generally includes a Diesel Oxidation Catalyst (DOC), a Diesel Particulate Filter (DPF) and a selective catalytic reaction (SCR) system. The control strategy on the aftertreatment system is the key for improving the emission of DHEVs. Lower exhaust temperature can reduce energy consumption, but too low a temperature will reduce the conversion efficiency of aftertreatment systems. Thus, the exhaust temperature of each component needs to be coordinated by a thermal management controller. In [23], a control method for the exhaust thermal management is proposed to coordinate the fuel economy and the emission performance of the aftertreatment system. The exhaust temperature of DHEVs controlled by a conventional power split strategy will drop when their engines are stopped, which likely leads to a considerable reduction in the conversion efficiency of their aftertreatment systems. In [24], a coordinated method controlling the post injection in-cylinder and the start of injection (SOI) is proposed for the power split and the exhaust thermal management of DHEVs. The influence of the weighting factor in the optimization objective on the fuel consumption and emissions is also discussed. In [25], a noncausal extended ECMS strategy for a DHEV equipped with a SCR system is proposed to minimize the exhaust emissions during cold start behavior. This strategy also proposes a three-state control framework including the battery energy, the exhaust temperature of the SCR system and the $NO_x$ emission, which also leads to an unstable co-state of the controller only working in a fixed time window. After the implementation of Euro VI regulations, there are higher requirements for controlling the exhaust temperature of aftertreatment systems [26]. Generally, separate fuel injection systems for rapidly raising the exhaust temperature are added to exhaust pipes, which increases the difficulties for the exhaust thermal management from engines to aftertreatment systems [27]. Therefore, it is highly necessary to propose a coordinated control method on the power split and the exhaust thermal management for fuel economy improvement and emissions reduction of DHEVs.

The road grade will change the power demand and operating conditions of DHEVs and seriously interfere with their control system, such that their fuel consumption and emissions will surge [28]. Vehicle controllers must actively adapt to road conditions to improve their performance [29]. Thanks to intelligent cruise systems communicating with the Global Positioning System (GPS) and Geographical Information System (GIS), road grade information can be obtained in advance, which can further improve the fuel consumption and emissions of DHEVs in real-time driving environments [30]. In [31], the authors regard the road grade as a Markov-chain model, and propose a stochastic MPC strategy for the power split to minimize fuel consumption and the variance of the battery *SoC*. In [32], the authors utilize preview information about the road grade to optimize a shifting schedule of an automatic transmission within MPC framework, such that the fuel consumption and driving performance can be improved. In [33], the authors obtain an optimal velocity trajectory under a given road grade profile, and solve the problem of minimizing fuel consumption. Literature [34] discusses the relationship between the $NO_x$ conversion goal and the road grade with its preview information for DHEVs. How to introduce road preview information into the control system of DHEVs to improve fuel consumption and emissions remains a further challenge.

In this paper, the main merits of the design procedure can be summarized in the following points. (i) The simulation model for a DHEV is developed for integrated optimization of power split and exhaust thermal management. (ii) The control method on the power split combining the rule-based and the optimization-based methods is designed to balance computational burden, computational accuracy, and anti-interference. (iii) To simultaneously achieve high fuel efficiency and low emissions, the hierarchical MPC framework coordinating the power split and the exhaust thermal management is proposed. (iv) To counteract interference from the road grade to the control system, road grade preview information is introduced into the solving process of the power split to improve fuel consumption and emissions.

To give a better explanation of the proposed model and the optimized methods, the remainder of this paper is organized as follows. Section 1 describes the DHEV model. Section 2 explains the problems associated with the rule-based method and the conventional optimization-based method for the power split through the simulation experiments. Section 3 presents two power splits combining the rule-based and the optimization-based methods. Section 4 presents two optimization-based methods for coordinating the power split and the exhaust thermal management and compares their control effectiveness. Section 5 introduces the road preview information in the power split strategy and depicts its effects on fuel consumption and emissions through the simulation experiments. Section 6 exhibits the conclusions.

## 2. DHEV Model

In this section, a physics based DHEV model with a hybrid structure including the problem statement of power split and exhaust thermal management is described, as shown in Figure 1. Among them, the power split and exhaust thermal loops are divided into the power loop and the exhaust thermal loop. In the power loop, the total demanded traction power ($P_d$) is provided via a power split device (PSD) that blends the engine output power ($P_e$) and the battery output power ($P_{bat} = P_m + P_g - P_{ch}$). Among these, $P_m$ is the power demand of motor, $P_g$ is the power demand of generator and $P_{ch}$ is the power demand of the battery charging. In the exhaust thermal loop, the exhaust temperature ($T_{ex}$) is one resource to provide heat power, and the separate fuel injection ($m_{post}$) that burns to release heat is another resource.

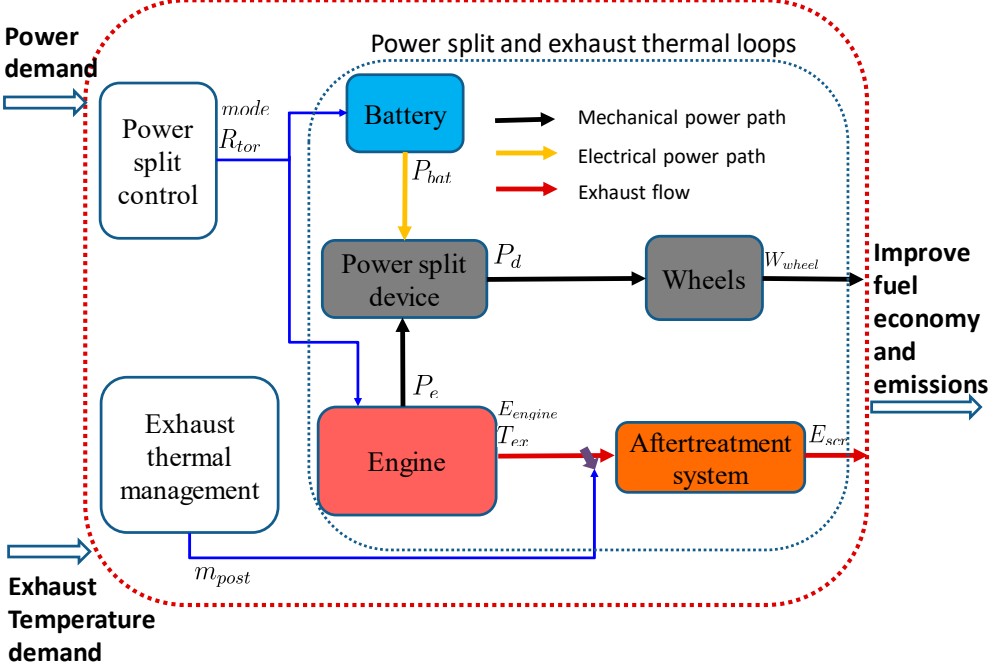

**Figure 1.** Schematic of the problem statement of the power split and exhaust thermal management.

Details of the DHEV model combining the mechanism and data are elaborated and shown in Figure 2. The model consists of two main parts: a hybrid vehicle model and a hybrid controller. The hybrid vehicle model includes a planetary gear set and vehicle module, an engine and emission aftertreatment system and exhaust thermal management module, an electric motor and generator module, and a battery $SoC$ module. The hybrid controller includes a driving control module and a brake regen control module. For tracking the profile of a driving cycle, the driver model utilizing a PID feedback controller obtains $P_d$ and the power demand of braking force ($P_b$), which are transmitted to the driving control module and the brake regen control module, respectively. The main function of the driving control module is to obtain the engine fuel injection command ($fuel^{cmd}$), the motor torque demand ($\mathcal{T}_m^{cmd}$) and the engine torque demand ($u_1$), based on signals including $P_d$, $SoC$, $P_{ch}$, the actual vehicle speed ($W_{wheel}$), the engine speed ($W_{engine}$), the generator speed ($W_{generator}$), and the motor speed ($W_{motor}$). The main function of the brake regen control module is to obtain the braking torque ($\mathcal{T}_{Brake}$) and the generator torque demand ($\mathcal{T}_g^{brake}$), based on signals including $P_b$, $SoC$, $W_{wheel}$ and $W_{generator}$. The main function of the engine and emission aftertreatment system and exhaust thermal management module is to obtain the engine torque ($\mathcal{T}_e$) and the SCR system emissions ($E_{scr}$), based on signals including $P_b$, $SoC$, $W_{wheel}$ and $W_{generator}$. The main function of the generator module is to obtain $P_g$ and the generator torque ($\mathcal{T}_g$), based on signals including $\mathcal{T}_g^{cmd}$ and $\mathcal{T}_g^{brake}$, etc. The main function of the electric motor module is to obtain $P_m$ and the motor torque ($\mathcal{T}_m$), based on $W_{motor}$ and $\mathcal{T}_m^{cmd}$. The main function of the Battery $SoC$ module is to obtain $SoC$ and $P_{ch}$, based on $P_g$ and $P_m$. The nomenclature of the constants used in the DHEV modeling are shown in Table 1.

**Table 1.** Constants nomenclature of the DHEV vehicle model.

| Symbol | Description | Value (Unit) |
|--------|-------------|--------------|
| $M$ | Vehicle mass | 2500 (kg) |
| $A_f$ | Face area | 2.52 (m$^2$) |
| $r$ | Dynamic tire radius | 0.51 (m) |
| $c_{p,EG}$ | Specific heat at constant pressure of the exhaust gas | $1 \times 10^3$ (J/kgK) |
| $c_{p,c}$ | Specific heat of the catalysts | 996 (J/kgK) |
| $m_{doc}$ | Mass of the catalytic converter of DOC | 19 (kg) |
| $m_{dpf}$ | Mass of the catalytic converter of DPF | 9.3 (kg) |
| $m_{scr}$ | Mass of the catalytic converter of SCR | 5.2 (kg) |
| $\varepsilon_{rad,doc}$ | Radiation coefficient of silencer of DOC | 0.61 (-) |
| $\varepsilon_{rad,dpf}$ | Radiation coefficient of silencer of DPF | 0.49 (-) |
| $\varepsilon_{rad,scr}$ | Radiation coefficient of silencer of SCR | 0.557 (-) |
| $\sigma_{sb}$ | Radiation constant | $5.07 \times 10^8$ (-) |
| $A_{rad,doc}$ | Radiating surface area of the silencer of DOC | 0.226 (m$^2$) |
| $A_{rad,dpf}$ | Radiating surface area of the silencer of DPF | 0.452 (m$^2$) |
| $A_{rad,scr}$ | Radiating surface area of the silencer of SCR | 1 (m$^2$) |

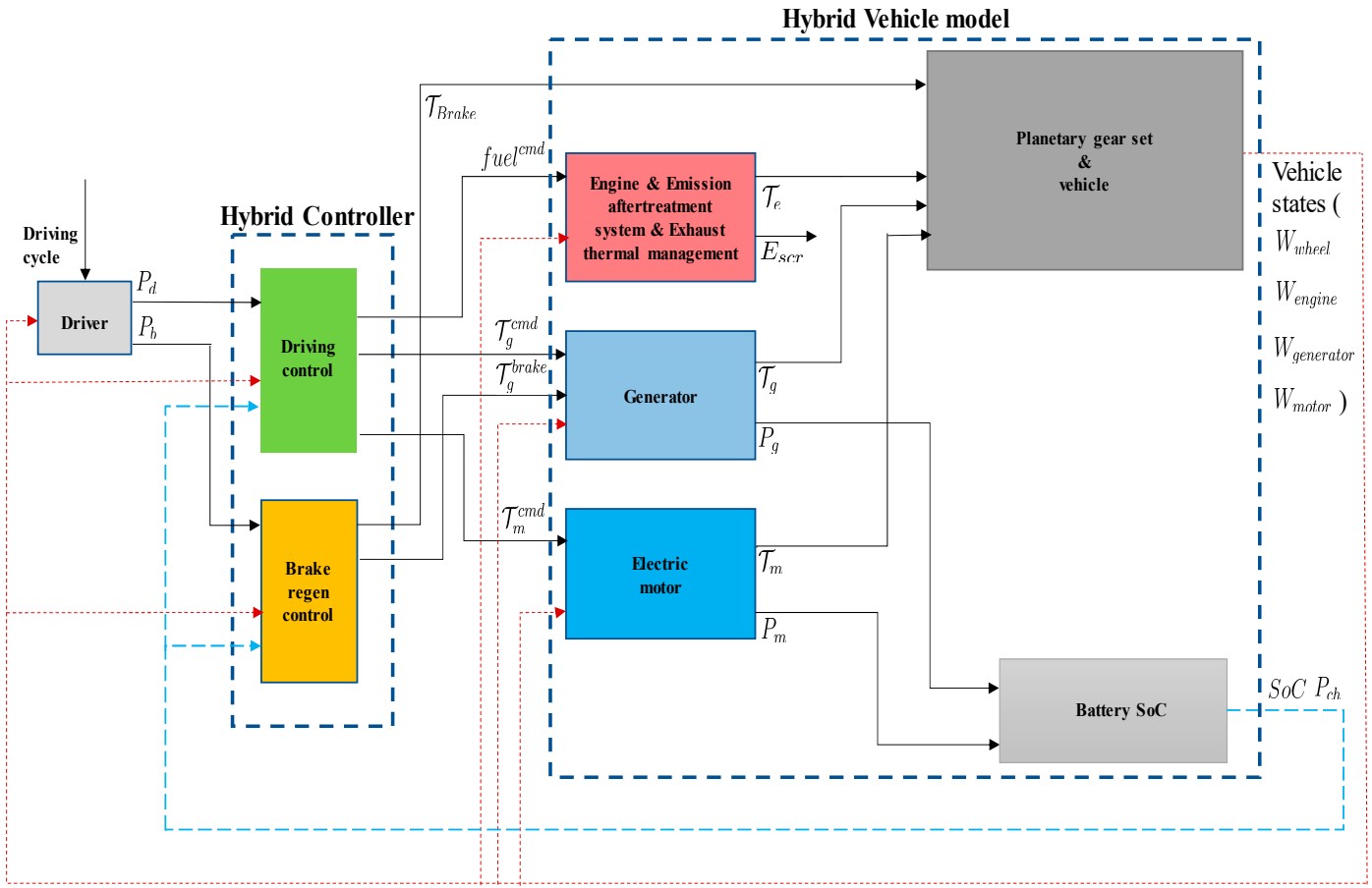

**Figure 2.** Schematic of the module composition and signal of the DHEV.

### 2.1. Planetary Gear Set and Vehicle Model

The vehicle longitudinal dynamics model presented here refers to the assumptions and simplifications in [35]. The vehicle gearbox utilizes a planetary gear structure. Its parameter setting and coupling relationship map refer to [36,37].

### 2.2. Exhaust Thermal Management Model

As shown in Figure 3, the engine and emission aftertreatment system and exhaust thermal management model mainly includes a diesel engine module, a DOC module, a DPF module, a SCR module and an exhaust thermal management module. The exhaust thermal management module utilizes a separate fuel injection system added to the exhaust pipe to control the exhaust temperature. Its main function is to obtain $m_{post}$, based on the exhaust mass flow ($m_{EG}^*$), $T_{ex}$ and the exhaust temperature of DOC ($T_{doc}$). The exhaust temperature and emissions signals are transmitted in cascade between the engine module, the DOC module, the DPF module and the SCR module. These signals also include the original engine emissions ($E_{engine}$), $T_{ex}$, the DOC emissions ($E_{doc}$), $T_{doc}$, the DPF emissions ($E_{dpf}$), the exhaust temperature of DPF ($T_{dpf}$) and the signal of SCR emissions ($E_{scr}$).

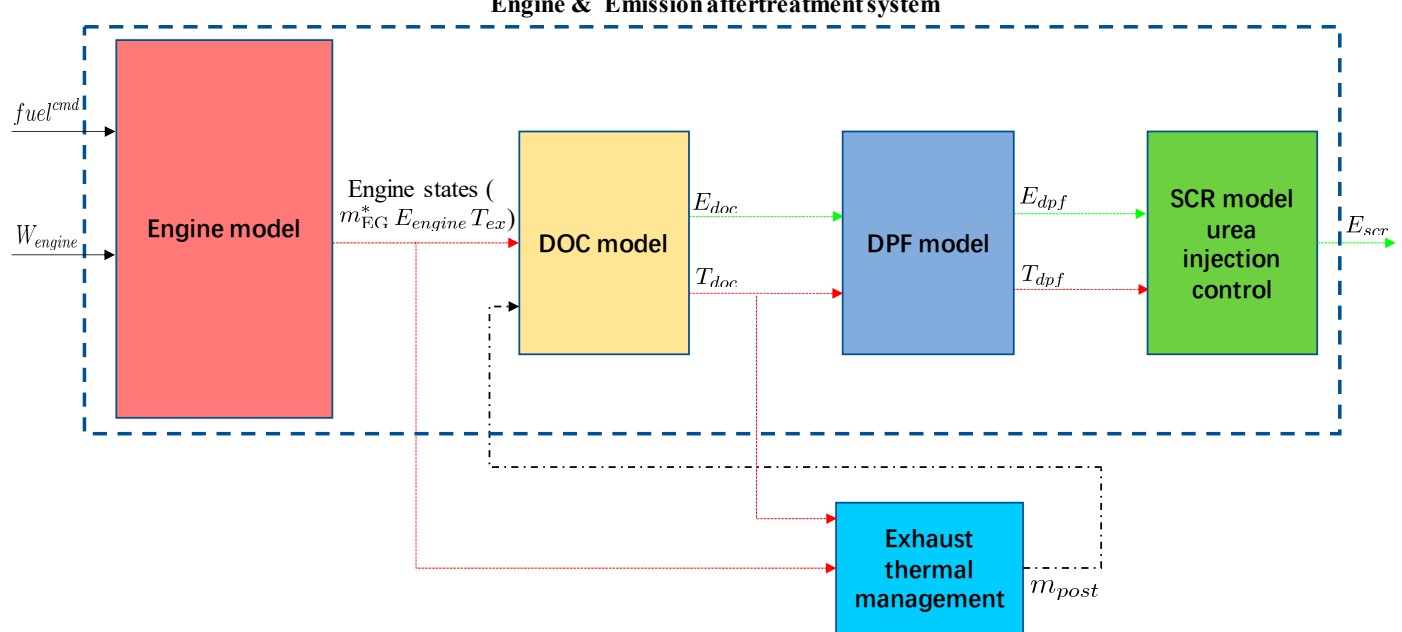

**Figure 3.** Schematic of the engine and emission aftertreatment system and exhaust thermal management model.

In this sub-section, an engine mean model is established first, with reference to the open source model on diesel engines proposed by the group of Professor Eriksson Lars [38,39]. Based on a diesel engine and its open-source dataset under the European Transient Cycle (ETC) [40], the parameters of the engine model are adjusted and identified. The diesel engine is manufactured by Changchun FAW Sihuan Engine Manufacture Co., Ltd. (Changchun, China). The engine has four cylinders and a total displacement volume of 2.771 L with a compression ratio (17.2:1), and it is turbocharged. Moreover, the engine incorporates an intercooler, its maximum speed is 3600 rpm, and its maximum torque is 350 Nm. The external characteristics of the engine are shown in Figure 4. Diesel engines are dynamic systems with complex nonlinear sub-modules. In this sub-section, a single injection method in the cylinder is employed to focus on the fuel consumption and emissions of the engine.

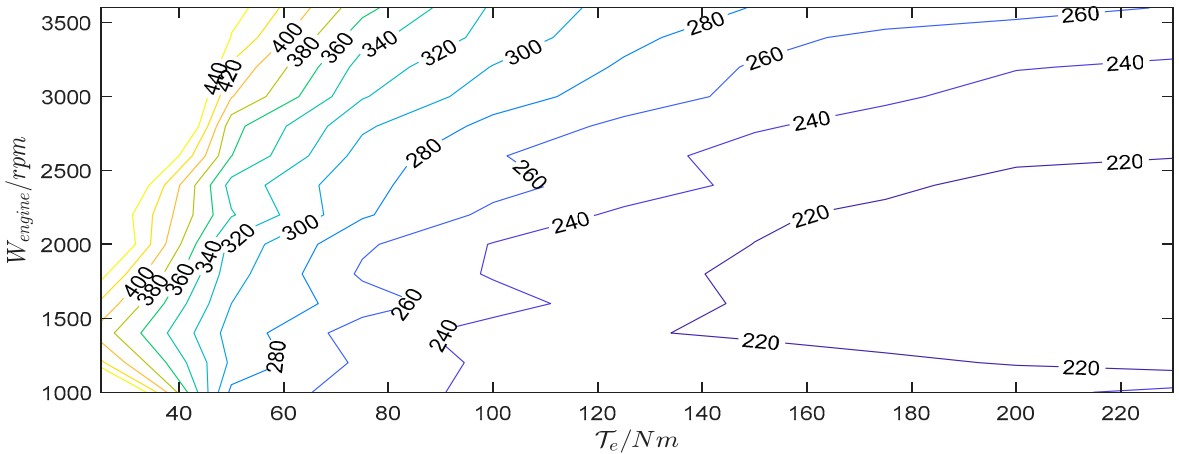

**Figure 4.** External characteristics of the CA4D28C5 diesel engine. The solid lines represent equal fuel consumption (g/kwh) contours.

In this sub-section, the data of the first 600 s part in the ETC cycle is used for parameter identification, and the data of the next 1200 s part in the ETC cycle is used for model

verification. The verification result is shown in Figure 5, the black lines in Figure 5(1–4) are the open-source data of the engine [40], and the red dotted lines in Figure 5(4) is the data of the model. Under the transient test conditions, the average error of fuel between the data and the model is 4.8%. That can satisfy the test requirements of the control system.

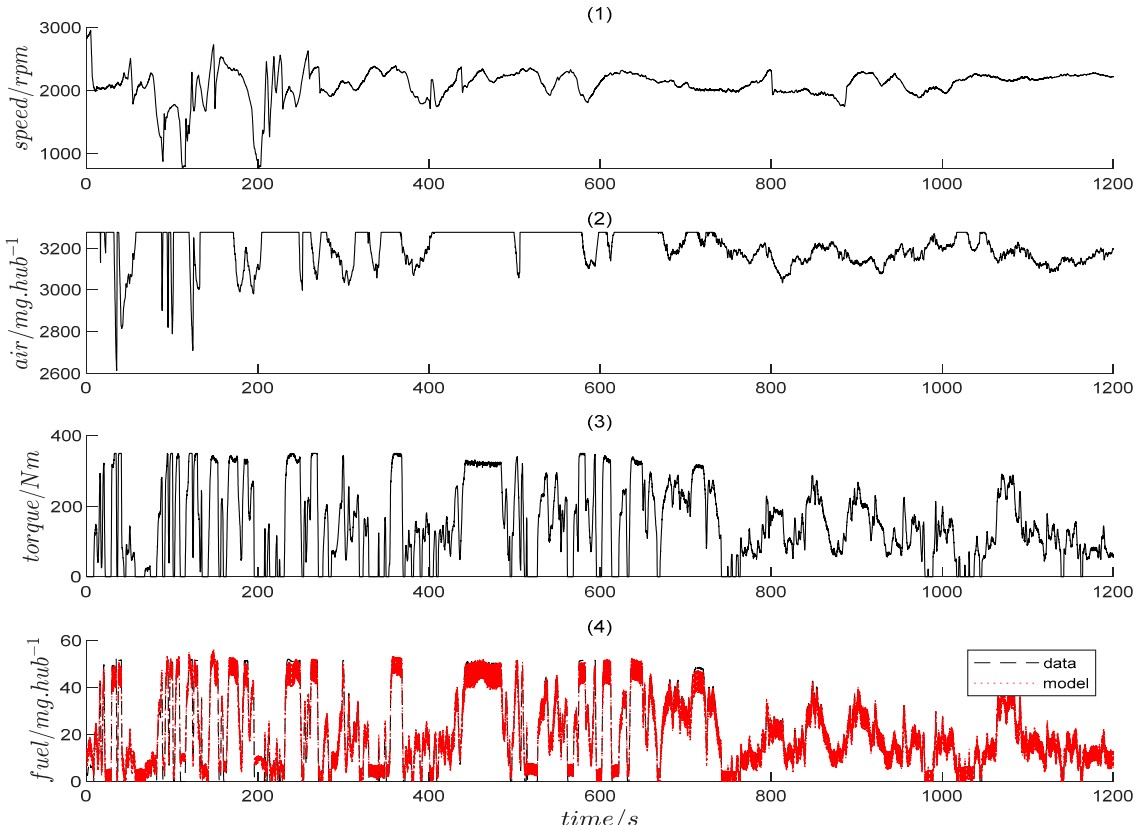

**Figure 5.** Verification result of the engine model under transient operating conditions: (**1**) engine speed, (**2**) air intake mass, (**3**) engine torque, (**4**) fuel consumption of engine.

A control-oriented model with a cascaded exhaust temperature dynamic for the DOC, DPF and SCR systems is established. As shown in Equation (1), the temperature model is three orders including $T_{doc}$, $T_{dpf}$ and $T_{scr}$. It also includes a model of the separate fuel injection system added to the exhaust pipe in the temperature dynamic of $T_{doc}$. The nomenclature of the constants used in the modeling are shown in Table 1. In addition, the relationship between the internal temperature and conversion efficiency of the SCR system can be described by a map [24]. As shown in Figure 6, the efficiency factor $CR_{NO_x}$ is calibrated through testing data from bench.

$$
\begin{aligned}
\dot{T}_{doc} &= \frac{c_{p,EG}m^*_{EG}(T_{ex}-T_{doc})}{c_{p,c}m_{doc}} - \frac{\varepsilon_{rad,doc}\sigma_{sb}A_{rad,doc}(T^4_{doc}-T^4_{amb})}{c_{p,c}m_{doc}} + \frac{m_{post}q_{lv}\eta}{c_{p,c}m_{doc}}, \\
\dot{T}_{dpf} &= \frac{c_{p,EG}m^*_{EG}(T_{doc}-T_{dpf})}{c_{p,c}m_{dpf}} - \frac{\varepsilon_{rad,dpf}\sigma_{sb}A_{rad,dpf}(T^4_{dpf}-T^4_{amb})}{c_{p,c}m_{dpf}}, \\
\dot{T}_{scr} &= \frac{c_{p,EG}m^*_{EG}(T_{dpf}-T_{scr})}{c_{p,c}m_{scr}} - \frac{\varepsilon_{rad,scr}\sigma_{sb}A_{rad,scr}(T^4_{scr}-T^4_{amb})}{c_{p,c}m_{scr}}.
\end{aligned}
\tag{1}
$$

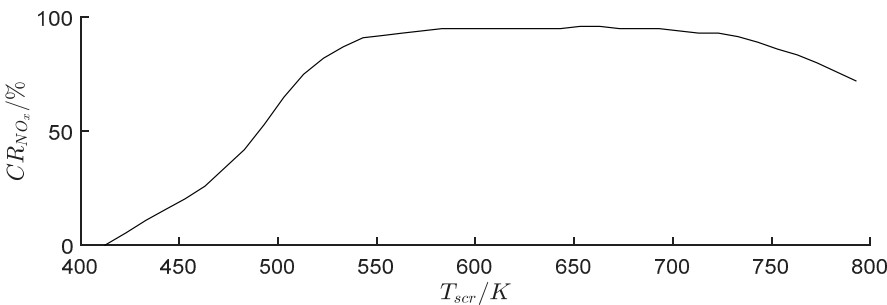

**Figure 6.** Effect of the temperature on the conversion efficiency of the SCR system.

### 2.3. Electric Motor and Generator Model

Figure 7 shows a power map of an electric motor. The working range of $W_{motor}$ is $[-6000, 6000]$ rpm, The working range of $\mathcal{T}_m$ is $[-150, 150]$ Nm. In this sub-section, the motor power is modeled as a function of $\mathcal{T}_m$ and $W_{motor}$. The analytical function is derived from the power map, and is fitted by a two-dimensional polynomial, as:

$$P_m = f_m(W_{motor}, \mathcal{T}_m) = a_1 W_{motor}^2 + a_2 \mathcal{T}_m^2 + a_3 W_{motor}\mathcal{T}_m + a_4 W_{motor} + a_5 \mathcal{T}_m + a_6, \quad (2)$$

where $a_m$, $m = 1\ldots6$ are tunable parameters and their values are shown in Table 2. In addition, a generator model the same as the motor model is used.

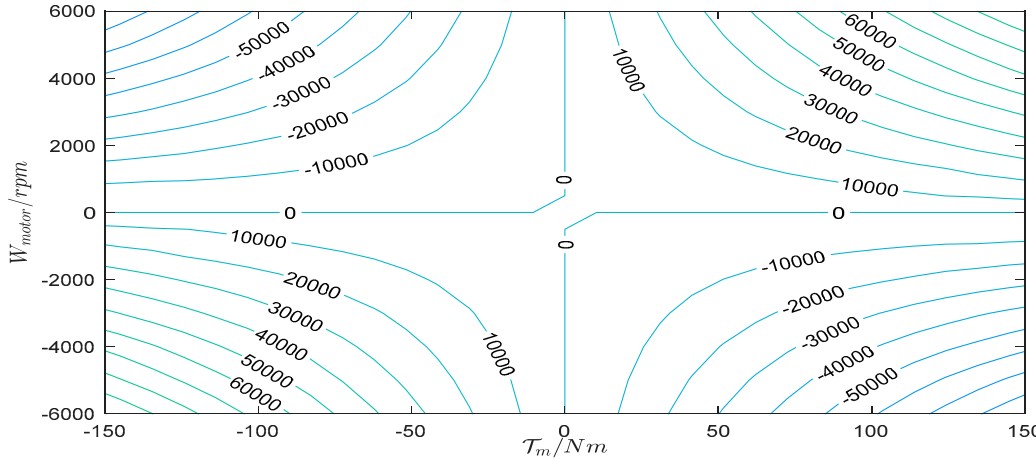

**Figure 7.** Equal power contours of an electric motor. The solid lines represent $P_m(W)$ based on the dimensional polynomial by fitting.

**Table 2.** Tunable parameters and values of the motor model.

| Symbol | $a_1$ | $a_2$ | $a_3$ | $a_4$ | $a_5$ | $a_6$ |
|--------|-------|-------|-------|-------|-------|-------|
| Value | 0.006556 | 0.2486 | 1.004 | $1.985 \times 10^8$ | $2.481 \times 10^7$ | $-2030$ |

### 2.4. Battery SoC Model

The battery performance (e.g., voltage ($V_{oc}$), internal resistance ($R_{int}$), current ($I$), and efficiency) is the outcome of thermally dependent electrochemical processes that are relatively complicated. Under the assumption that battery states are temperature independent, the current and the battery *SoC* reflecting the energy states can be expressed as Equation (3), where $C_{batt}(100, 90 \text{ mAh})$ is the nominal battery capacity.

$$
\begin{aligned}
I(t) &= \frac{V_{oc}(t) - \sqrt{V_{oc}^2(t) - 4R_{int}(t)P_m(t)}}{2R_{int}(t)}, \\
\dot{SoC}(t) &= -\frac{I(t)}{C_{batt}}.
\end{aligned}
\quad (3)
$$

*2.5. Hybrid Controller Model*

As shown in Figure 8, the driving control model mainly includes a power split module and a torque optimal module for engine, motor and generator. The main function of the power split module is to calculate the hybrid operating mode (*mode*) and $P_e$ based on certain rules, in accordance with $P_d$, *SoC*, $P_{ch}$ and $W_{wheel}$. The main function of the engine motor generator torque optimal module is to obtain $fuel^{cmd}$, $\mathcal{T}_m^{cmd}$ and $\mathcal{T}_g^{cmd}$ based on certain rules.

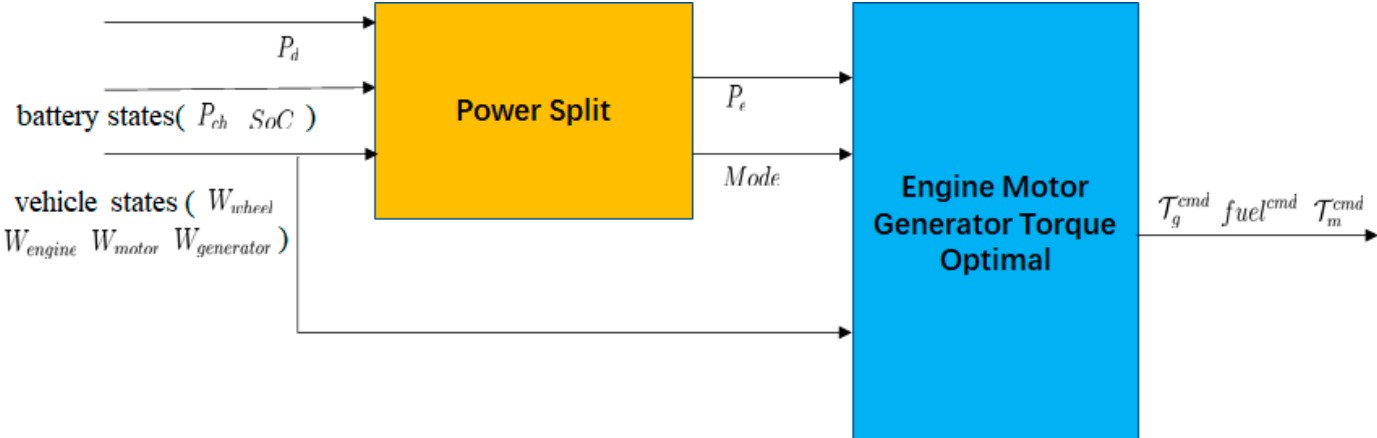

**Figure 8.** Schematic of the Driving control model.

In the battery controller, the maximum *SoC* is 0.8 and the minimum *SoC* is 0.5. The maximum charge and discharge power of the battery is 24,000 W. When $W_{wheel} > 0.5$ rad/s and *SoC* $< 0.8$, the braking energy recovery starts under the maximum torque limit corresponding to the current generator speed. In addition, the excess braking force is provided by the braking system.

*2.6. General Problem Statement of the Power Split and Exhaust Thermal Management*

The key to control DHEVs is the power split between the engine and the motor for coordinating fuel consumption and emissions. The exhaust temperature of DHEVs will drop when their engines are stopped. Thus, the exhaust temperature should be managed within a specified range, otherwise the efficiency of the aftertreatment system will become deteriorative. As shown in Figure 1, the states of the controller for DHEVs that may be of interest are *SoC* and exhaust temperature. The control inputs may consist of *mode*, $R_{tor}$ and $m_{post}$. Due to the fuel consumption sensitivity to the separate fuel injection system, the coordinated exhaust thermal management between engine exhaust and the separate fuel injection should be careful. Overall, the main objective of the power split and exhaust thermal management is to minimize the fuel consumption and emissions while enforcing the power and thermal constraints.

## 3. Two Power Split Method and Their Evaluation

In this section, a rule-based controller (RC) and a conventional optimization-based controller for the power split are designed, respectively. Their control effectiveness is evaluated through the simulation experiments.

*3.1. The Rule-Based Power Split Method and Its Evaluation*

The rule-based controller (RC) for the power split is designed in accordance with the characteristics of the battery, the motor and the engine. Its control rules are as follows:

- When *SoC* $> 0.5$ (the battery energy is sufficient), $P_d < P_{ev}(12,000$ W$)$ and $W_{wheel} < 26$ rad/s, the working mode of the power split is the motor mode, and $P_m = P_d$.

- When $P_{ev} < P_d < P_{emax}(32,000 \text{ W})$, the working mode of the power split is the engine mode, and $P_e = P_d + P_{ch}$.
- When $P_d > P_{emax}$ and $SoC > 0.5$, the working mode of the power split is the hybrid mode, and $P_e = P_{emax}$ and $P_m = P_d - P_e$.

Remark: Considering its characteristics of fuel consumption, the switching threshold of the engine power from the engine mode to the hybrid mode is set to 32,000 W in order to satisfy the power requirements of the velocity trajectory tracking under the WLTC cycle and maximize the advantages of motor assist. In addition, when $SoC < 0.5$, $P_m = 0$. When $SoC < 0.63$, $P_{ch} > 0$.

Firstly, the RC for the power split is tested under the WLTC cycle, its transient working process is shown in Figure 9. The motor, engine and hybrid operating modes are continuously switched during the transient WLTC cycle as shown in Figure 9(3). As shown in Figure 9(4), during conditions of lower speed (about 0–700 s), the instantaneous fuel consumption of the engine was 200–250 g/kwh. During conditions of higher speed (about 700–1800 s), the instantaneous fuel consumption was approximately 250 g/kwh because of the influence of the vehicle speed. In the 0–130 s interval, the vehicle is mainly driven by the motor, and $T_{doc}$ dropped by approximately 100 K from the initial 550 K (warm-up condition). Although $T_{scr}$ drops with a certain delay, the conversion efficiency of the SCR system obviously drops after the 200th second. As the engine of the hybrid system works intermittently, it may be difficult to ensure the ideal operating temperature (generally more than 500 K) for the aftertreatment systems, resulting in a reduced conversion efficiency of emissions. Therefore, it is necessary for DHEVs to properly consider the exhaust temperature during the power split stage.

### 3.2. The Optimization-Based Power Split Method and Its Evaluation

This sub-section designs a mode and power optimized controller (MPOC) for the power split and exhaust thermal management, with reference to the optimization-based power split strategy proposed in [41]. The optimization problem of this control strategy is described in Equation (4). The states are selected as $SoC$ and $T_{doc}$, and the control inputs are *mode* and $P_{bat}$.

$$
\begin{aligned}
\min J_{mpoc} &= \int_t^{t+T} \tau_{fuel}^1 W_{fuel}(t') + \tau_i^1 I(t') + \tau_{Tdoc}^1 e_{doc} u(e_{doc})(t') dt', \\
s.t. \dot{T}_{doc} &= \frac{c_{p,EG} m_{EG}^*(T_{ex} - T_{doc})}{c_{p,c} m_{doc}} - \frac{\varepsilon_{rad,doc} \sigma_{sb} A_{rad,doc}(T_{doc}^4 - T_{amb}^4)}{c_{p,c} m_{doc}}, \\
\dot{SoC}(t) &= -\frac{I(t)}{C_{batt}}, \\
I(t) &= \frac{V_{oc}(t) - \sqrt{V_{oc}^2(t) - 4R_{int}(t)P_{bat}(t)}}{2R_{int}(t)}, \\
P_d &= P_e + P_{bat}, \\
-24 \text{ kw} &< P_{bat} < 24 \text{ kw}, \\
0.5 &< SoC < 0.8, \\
0 &< T_{doc} < 800 \text{ K}, \\
\text{mod} e &\in \{1,2\}, \\
e_{doc} &= T_{doc,des} - T_{doc}, \\
u(e_{doc}) = 0 \quad &when \quad T_{doc} \geq T_{doc,des}, \\
u(e_{doc}) = 1 \quad &when \quad T_{doc} < T_{doc,des},
\end{aligned}
\tag{4}
$$

where the cost related to the fuel consumption $W_{fuel}$ is multiplied by a weight $\tau_{fuel}^1$, the battery current $I$ is multiplied by a weight $\tau_i^1$, and the difference between $T_{doc}$ and its target $T_{doc,des} = 600$ k is multiplied by a weight $\tau_{Tdoc}^1$.

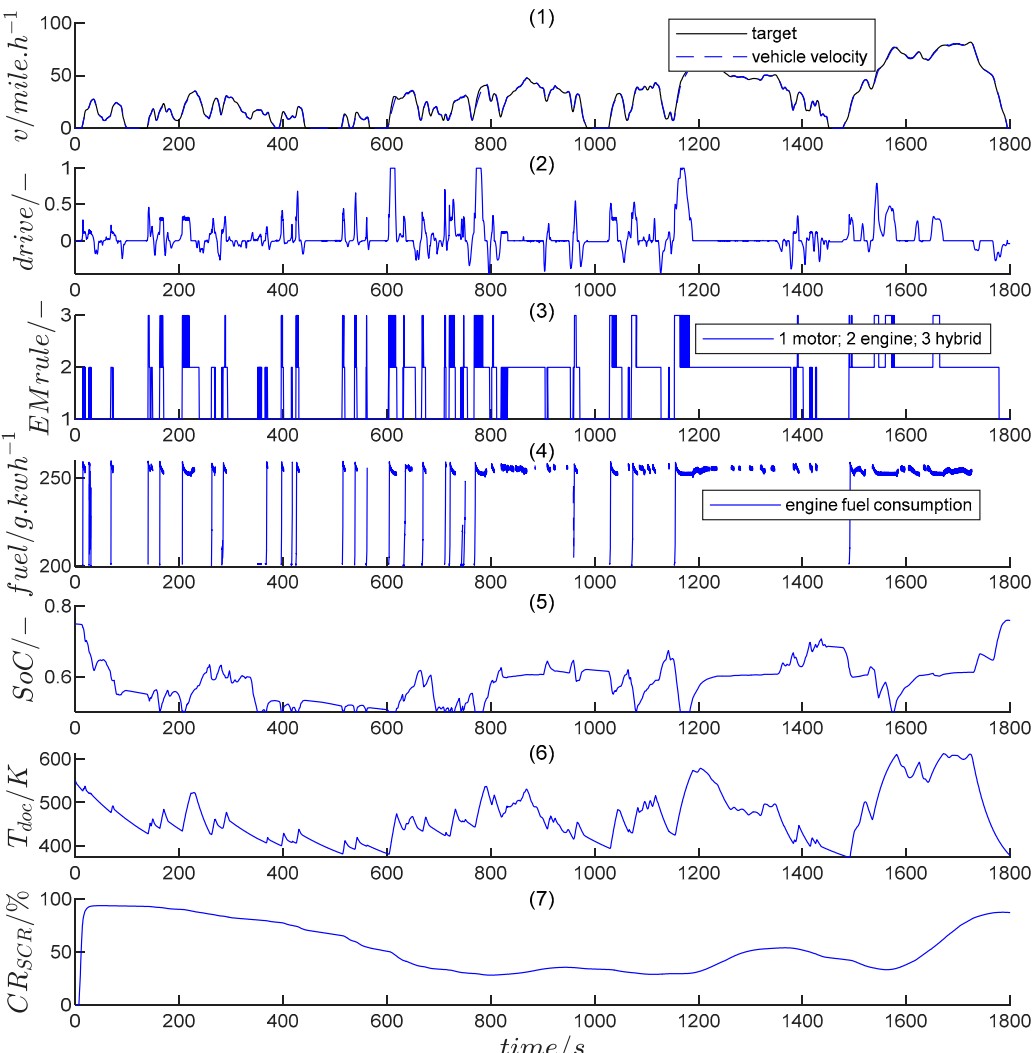

**Figure 9.** Working process of the diesel hybrid electric system under the RC method: (**1**) the speed target of the WLTC cycle and the tracking states of the controller, (**2**) the normalized curve of the braking force and driving force, (**3**) the switching process of energy mode rule (EM rule), 1 represents the motor mode, 2 represents the engine mode and 3 represents the hybrid mode, (**4**) the instantaneous fuel consumption of the engine, (**5**) $SoC$, (**6**) $T_{doc}$, (**7**) the NO$_x$ conversion efficiency in the SCR system.

The simulation is conducted on a desktop computer [42] with an Intel Core CPU (2.60 GHz) in MATLAB/SIMULINK. The sequence of control inputs was optimized by minimizing the proposed cost functions with respect to the constraints using the command fmincon with a SQP algorithm. The time step used in the NMPC controllers is 0.02 s, while the length of the prediction horizon is 10. The MPOC method is tested under the typical urban road conditions as shown in Figure 10. The computational time obtained using a MATLAB command is approximately 374 s for the entire process.

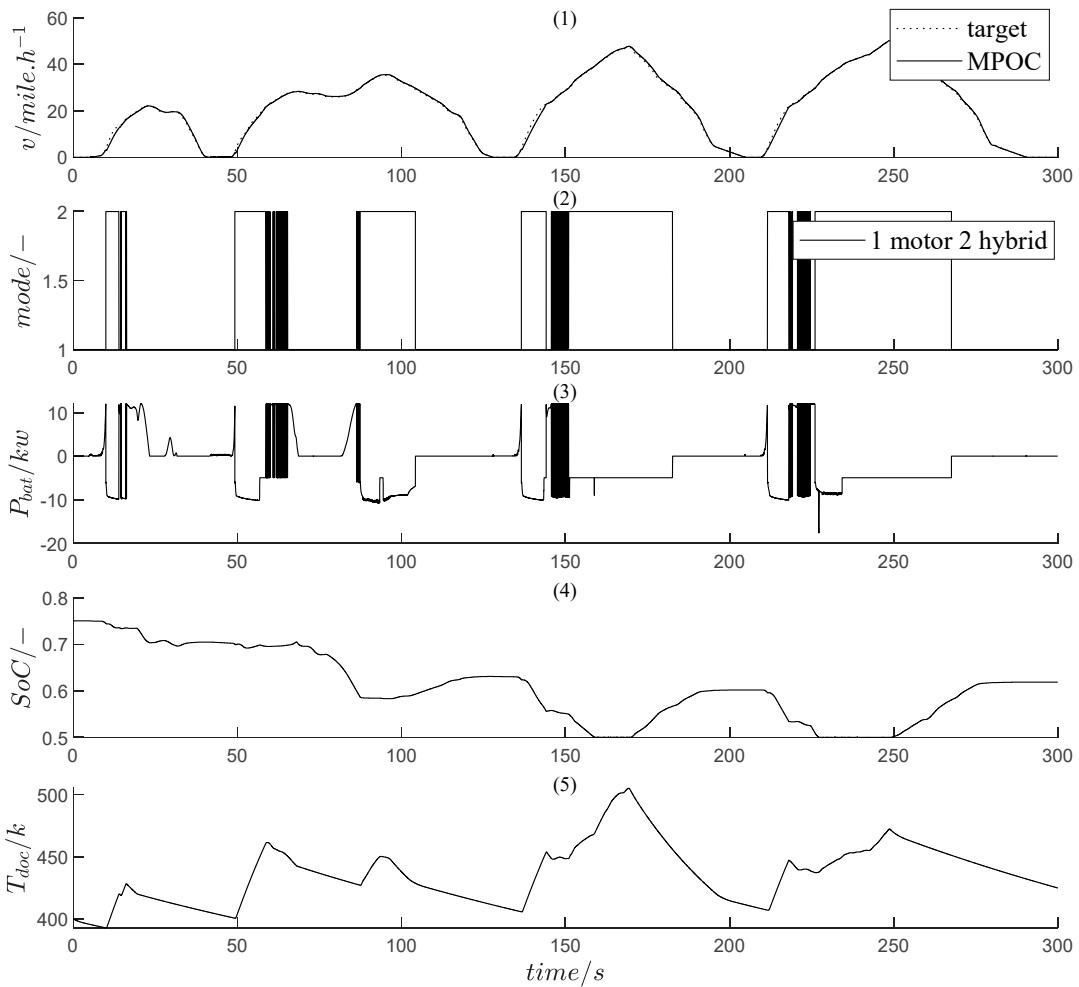

**Figure 10.** Working process of the diesel hybrid electric system under the MPOC method: (**1**) the speed target of the cycle and the tracking states of the controller, (**2**) the switching process of energy mode rule (EM rule), 1 represents the motor mode, 2 represents the engine mode, (**3**) $P_{bat}$, (**4**) *SoC*, (**5**) $T_{doc}$.

## 4. The Power Split Combining the Rule-Based and the Optimization-Based Methods and Their Comparison on Control Effectiveness

The rule-based RC method does not consider the exhaust temperature of engine, therefore they cannot adapt to all conditions. When the optimization-based MPOC method is utilized to solve the power split, the complex optimization problem with an integer variable (engine start or stop) and some constraints are introduced, which leads to a long computational time. Therefore, in order to coordinate the power split and exhaust thermal management and minimize the computational burden, this paper proposes several strategies combining the rule-based and the optimization-based methods under the MPC framework. These strategies retain the motor mode and engine mode in the rule-based strategy proposed in Section 2.1, and improve the hybrid mode by optimizing the power split factor ($R_{tor} = P_m/(P_m + P_e)$) between the engine and the motor.

### 4.1. Fuel Optimized Controller, FOC

Firstly, a fuel optimized controller (FOC) for power split is proposed, with the associated optimization problem described by Equation (5). The state is selected as *SoC*, and the control input is $R_{tor}$. In addition to the fuel consumption $W_{fuel}$, the battery current *I* is also used as an objective. This is because when the current is smaller, the battery power is

smaller, and when the current change is smaller, the additional energy loss is smaller, and the battery aging is reduced.

$$
\begin{aligned}
\min J_{foc} = & \int_t^{t+T} \tau_{fuel}^2 W_{fuel}(t') + \tau_i^2 I(t') dt', \\
& s.t. \dot{SoC}(t) = -\frac{I(t)}{C_{batt}}, \\
& I(t) = \frac{V_{oc}(t) - \sqrt{V_{oc}^2(t) - 4R_{int}(t)P_m(t)}}{2R_{int}(t)}, \\
& P_m = P_d R_{tor}, \\
& P_{Eeg} = P_d(1 - R_{tor}),
\end{aligned}
\tag{5}
$$

where the cost related to $W_{fuel}$ is multiplied by a weight $\tau_{fuel}^2$ and $I$ is multiplied by a weight $\tau_i^2$. The constraints are $0 < R_{tor} < 1$ and $0.5 < SoC < 0.8$.

### 4.2. Fuel and Thermal Optimized Controller, FTOC

In this sub-section, a separate fuel injection system for rapidly raising the exhaust temperature is added to the exhaust pipe. The normal operation of the fuel injection system requires the DOC system to satisfy a certain working temperature. The exhaust temperature is the only way to control $T_{doc}$. In the hybrid mode phase, the exhaust temperature can be controlled by adjusting $R_{tor}$. In the critical phase of the ideal working temperature of the DOC system, the slightly increased exhaust temperature by adjusting $R_{tor}$ can effectively improve the working efficiency of the fuel injection system. Therefore, a fuel and thermal optimized controller (FTOC) for power split simultaneously considering fuel consumption and exhaust temperature under multi-constraints is proposed. This method can control $T_{doc}$ in the hybrid mode phase. The optimization problem of FTOC is described as Equation (6). The states are selected as $SoC$ and $T_{doc}$, and the control input is $R_{tor}$.

$$
\begin{aligned}
\min J_{ftoc} = & \int_t^{t+T} \tau_{fuel}^3 W_{fuel}(t') + \tau_i^3 I(t') + \tau_{Tdoc}^3 e_{doc} u(e_{doc})(t') dt', \\
& s.t. \dot{T}_{doc} = \frac{c_{p,EG} m_{EG}^* (T_{ex} - T_{doc})}{c_{p,c} m_{doc}} - \frac{\varepsilon_{rad,doc} \sigma_{sb} A_{rad,doc} (T_{doc}^4 - T_{amb}^4)}{c_{p,c} m_{doc}}, \\
& \dot{SoC}(t) = -\frac{I(t)}{C_{batt}}, \\
& I(t) = \frac{V_{oc}(t) - \sqrt{V_{oc}^2(t) - 4R_{int}(t)P_{bat}(t)}}{2R_{int}(t)}, \\
& P_m = P_d R_{tor}, \\
& P_{Eeg} = P_d(1 - R_{tor}), \\
& 0 < R_{tor} < 1, \\
& 0.5 < SoC < 0.8, \\
& 0 < T_{doc} < 800 \text{ K}, \\
& e_{doc} = T_{doc,des} - T_{doc}, \\
& u(e_{doc}) = 0 \quad when \quad T_{doc} \geq T_{doc,des}, \\
& u(e_{doc}) = 1 \quad when \quad T_{doc} < T_{doc,des},
\end{aligned}
\tag{6}
$$

where the cost related to $W_{fuel}$ is multiplied by a weight $\tau_{fuel}^3$, $I$ is multiplied by a weight $\tau_i^3$ and difference between $T_{doc}$ and $T_{doc,des} = 600$ k is multiplied by a weight $\tau_{Tdoc}^3$.

### 4.3. Evaluation of the Controllers for Power Split

The comparison results between the RC, the FOC and the FTOC methods under typical urban road conditions are shown in Figure 11. As shown in Figure 11(1), there is little difference in the tracking effectiveness of the three controllers. The switching process of energy mode rule of the RC is shown in Figure 11(2), and the average fuel consumption during the entire process is 12.61 mg/cycle. Curves of $R_{tor}$ of the FOC and the FTOC are shown in Figure 11(3,4), respectively. $R_{tor} = 1$ represents the motor mode, $R_{tor} = 0$ represents the engine mode and $R_{tor}(0-1)$ represents the hybrid mode. The average fuel consumption during the entire process of the FOC and the FTOC is 12.91 mg/cycle and 12.37 mg/cycle, respectively. Comparison results of $SoC$ state trajectories among the three

controllers are shown in Figure 11(5). The terminal *SoC* states of the RC and the FTOC are almost the same, the terminal *SoC* value of the FOC is higher by 0.03 compared with the RC and the FTOC. As shown in Figure 11(6), the initial temperature of $T_{doc}$ is 400 K. There is little difference in the $T_{doc}$ curves of the RC and the FOC, and $T_{doc}$ takes nearly 90 s to reach the lowest operating temperature of the separate fuel injection system. During this process, the emission conversion efficiency will be very low so that severe emissions will be caused. Compared with the RC and the FOC, $T_{doc}$ of the FTOC takes nearly 60 s to reach the lowest operating temperature, which is nearly 30 s earlier and greatly improves the working efficiency of the aftertreatment system. As shown in Figure 11(2–4), starting from 50 s the $R_{tor}$ of the FTOC is significantly lower to increase the engine power output. At the same time, the motor output power of the FTOC is lower so that the battery *SoC* is significantly higher than that of the RC and the FOC during the next 100 s as shown in Figure 11(5). As the engine power output of the FTOC increases, its $T_{doc}$ rises to nearly 480 K in advance as shown in Figure 11(6).

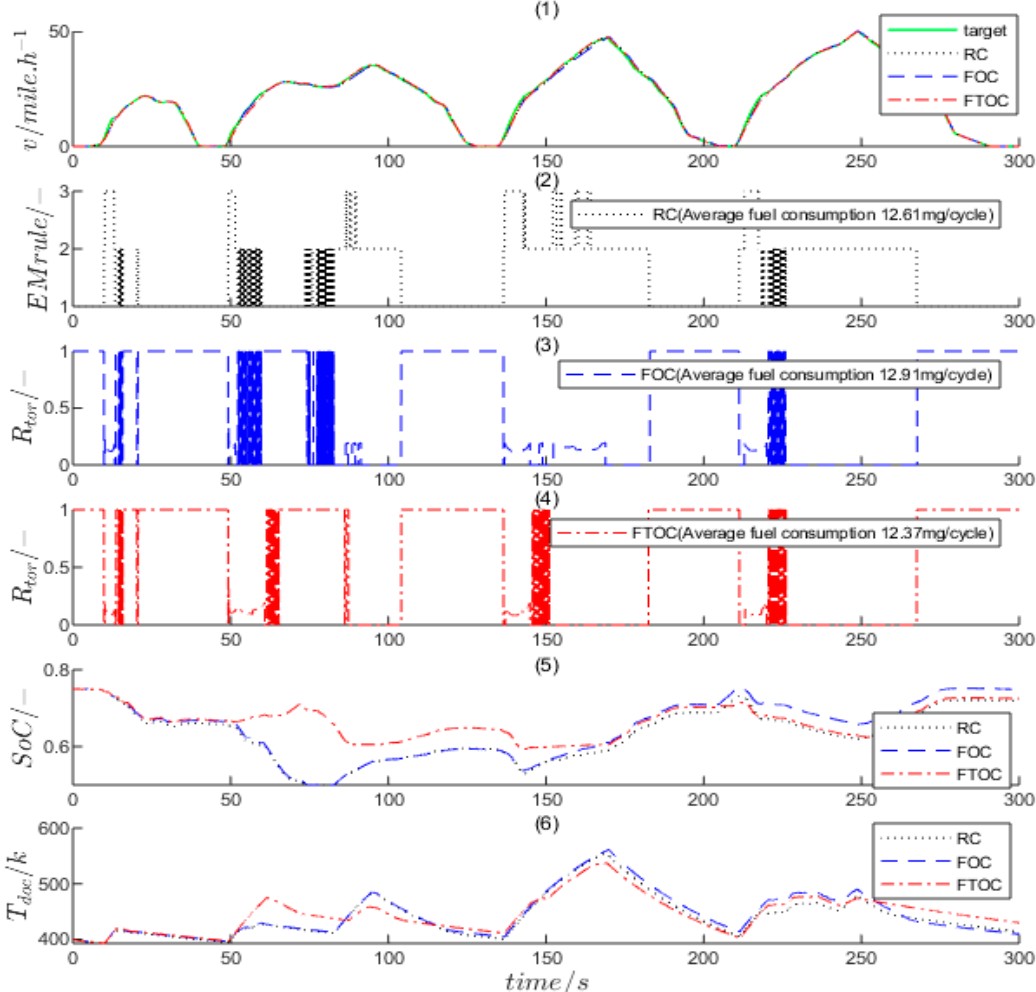

**Figure 11.** Comparison results of the RC vs. the FOC vs. the FTOC: (**1**) the speed target of the cycle and the tracking states of the controllers, (**2**) energy mode rule of the RC, (**3**) $R_{tor}$ of the FOC, (**4**) $R_{tor}$ of the FTOC, (**5**) *SoC*, (**6**) $T_{doc}$.

Next, the fuel consumptions of the three controllers are analyzed. Due to the differences in the *SoC* terminal states of the three controllers, the linear correction method [11], based on the charging/discharging characteristics of the hybrid system, is utilized to correct the fuel consumption. The comparison results of the average fuel consumption during the entire transient process show that the RC's is 13.61 (12.61 + 1) mg/cycle, that the FTOC's is

13.37 (12.37 + 1) mg/cycle, and that the FOC's is 12.91 mg/cycle, which saves nearly 5% compared with the RC. Overall, the FTOC proposed can not only increase $T_{doc}$ in advance, but also has a slightly lower fuel consumption than the RC. Moreover, the computational time of the FTOC is approximately 134 s for the process, which is reduced by nearly 60% compared with the MPOC.

## 5. Design and Evaluation of Two Optimization-Based Methods for Coordinating the Power Split and the Exhaust Thermal Management

In this section, a sequential control method and a hierarchical control method are proposed to coordinate the power split and the exhaust thermal management, and the simulations are conducted to invalidate their control effectiveness.

### 5.1. Overall Design Structure of the Two Controllers

In this sub-section, a fuel post-injection thermal controller (FPTC) which adjusts the separate fuel injection system for exhaust thermal management is introduced without affecting the control framework of the power split. Its control rules are as follows: When $T_{doc} \geq 440$ k and the engine is on, the FPTC starts to work. The objective temperature of $T_{doc}$ is 600 K by a PID controller. The FPTC is combined with the RC and the FTOC, respectively, and two control methods for the power split and the exhaust thermal management are proposed. Among them, the schematic of the sequential control method (RC + FPTC) is shown in Figure 12 and the schematic of the hierarchical control method (FTOC + FPTC) is shown in Figure 13. Compared with the control input of the RC, the FTOC introduces additional $T_{doc}$. From the power split to the exhaust thermal management, a hierarchical control of the exhaust temperature is realized by the FTOC + FPTC.

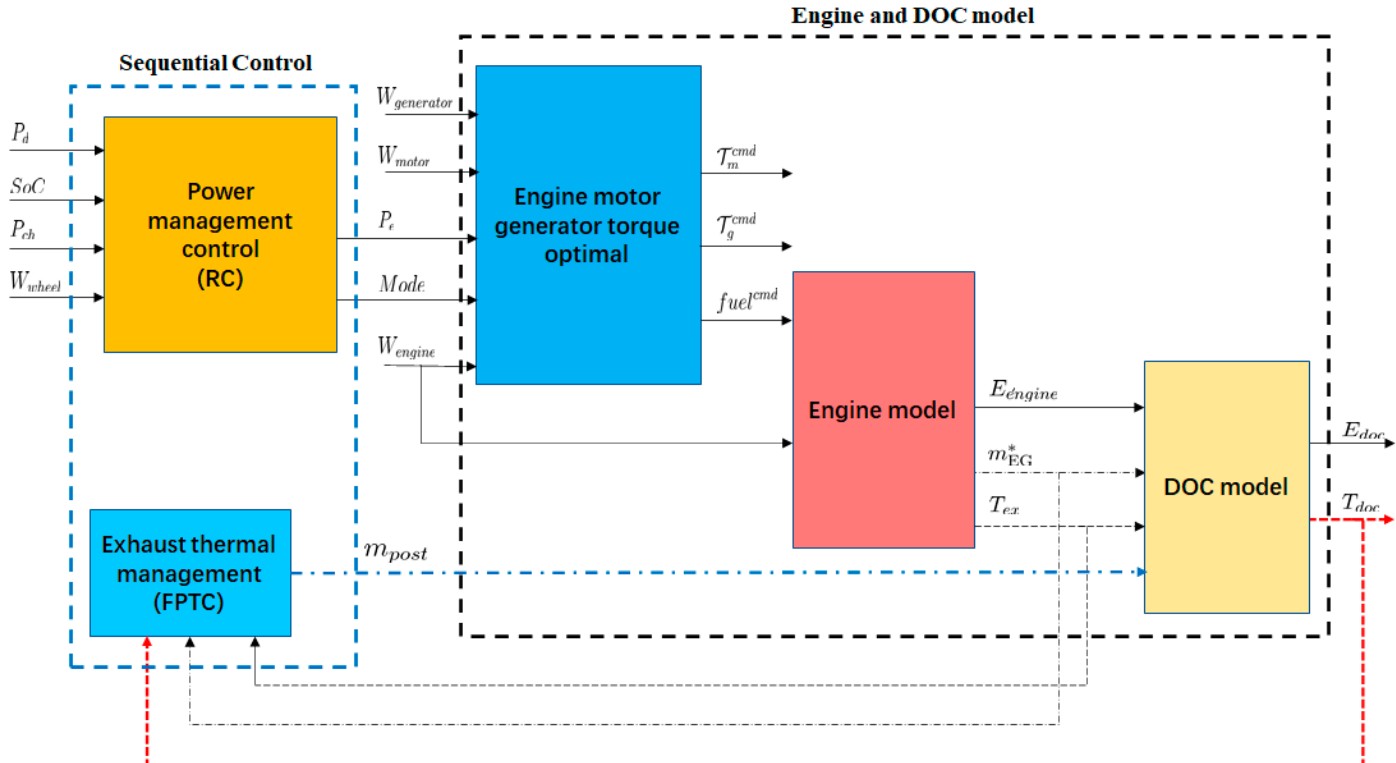

**Figure 12.** Schematic of sequential control for power split and exhaust thermal management (RC + FPTC).

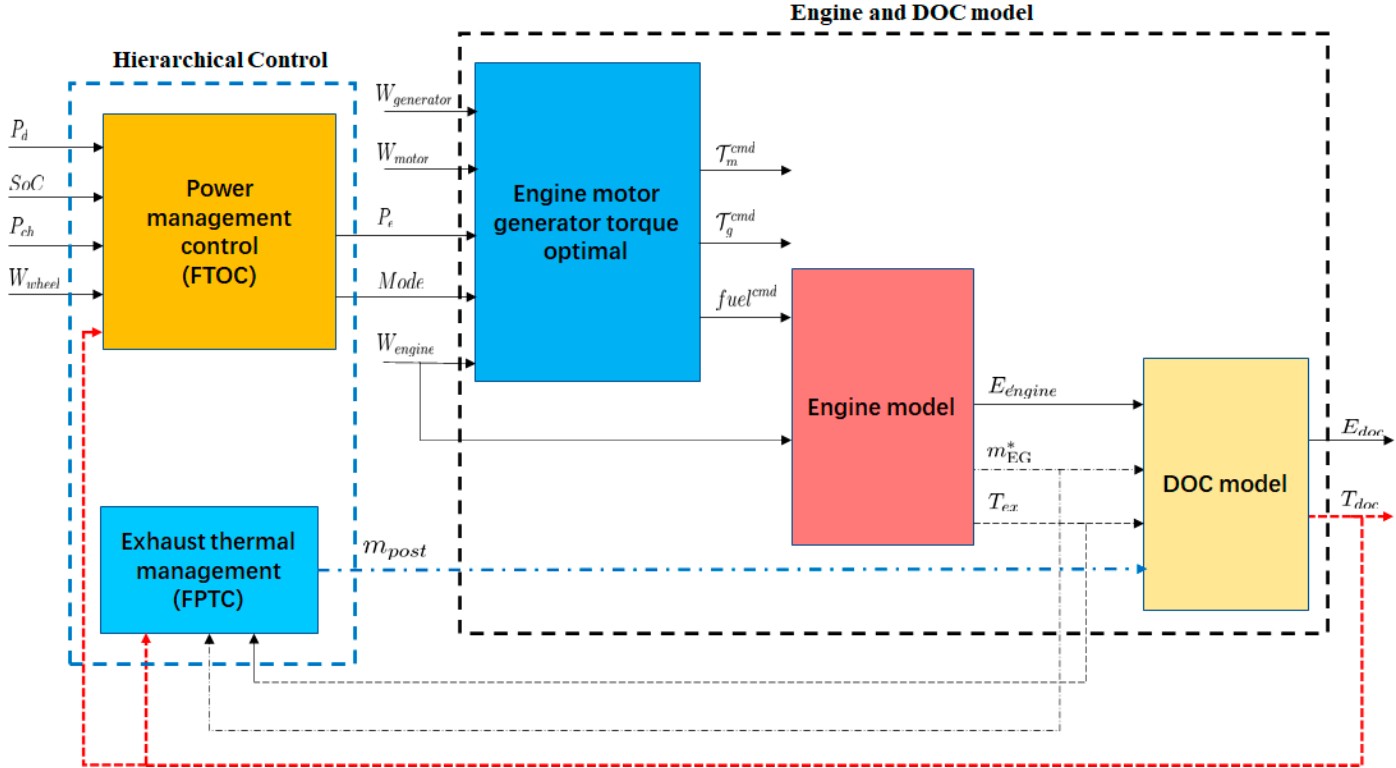

**Figure 13.** Schematic of hierarchical control for power split and exhaust thermal management (FTOC + FPTC).

### 5.2. Evaluation of the Two Controllers

Results compared between the RC + FPTC and the FTOC + FPTC methods under the WLTC cycle are shown in Figure 14. When the power split is the motor mode with engine off (such as at the 300–360 s and 420–510 s phase), the FPTC controller stops such that $T_{doc}$ drops. Apart from these phases, both controllers can maintain $T_{doc}$ at around 600 K.

Compared with the RC + FPTC around the 200th second, the FTOC + FPTC can increase $T_{doc}$ to above 600 K around the 160th second, which is nearly 40 s earlier at reaching the ideal temperature, and greatly improves the working efficiency of the aftertreatment system as shown in Figure 14(3). The terminal *SoC* value of the RC + FPTC is higher by 0.05 than that of the FTOC + FPTC as shown in Figure 14(4). At approximately the 150–220 s phase, the cumulative fuel consumption of the FTOC + FPTC is slightly higher than that of the RC + FPTC. This is because the FTOC can increase its $T_{doc}$ to 440 K in advance, which satisfies the lowest operating temperature of the FPTC. After that, the cumulative fuel consumption of the FTOC + FPTC is relatively lower than that of the RC + FPTC.

The comparison results of the total fuel consumption and the total $NO_x$ emission between the RC + FPTC and the FTOC + FPTC methods under the WLTC cycle are shown in Figure 15. The total fuel consumption (the injection in-cylinder plus the separate injection in exhaust pipe) of the RC + FPTC is 2463 g, and that of the FTOC + FPTC is 2194 g. To compare the energy consumption, the linear correction method [11] based on the charging/discharging characteristics of the hybrid system is utilized. If the terminal *SoC* value of the FTOC + FPTC is raised to the terminal value of the RC + FPTC, the fuel consumption needs to be increased by 50 g at most. Thus, the total fuel consumption of the FTOC + FPTC is 2244 g (2194 + 50 g), which is reduced by at least 8.9% compared with the RC + FPTC's 2463 g. In addition, the total $NO_x$ emission of the FTOC + FPTC is 284 g, which is reduced by 10% compared with the RC + FPTC's 316 g. The improvements in the fuel consumption and the $NO_x$ emission are almost the same, this is because the hybrid system does not adopt a separate optimization strategy for the $NO_x$ emission. Overall, compared with the RC + FPTC, the FTOC + FPTC can reach the optimal working temperature of

the aftertreatment system in advance, and its total fuel consumption and $NO_x$ emission are lower.

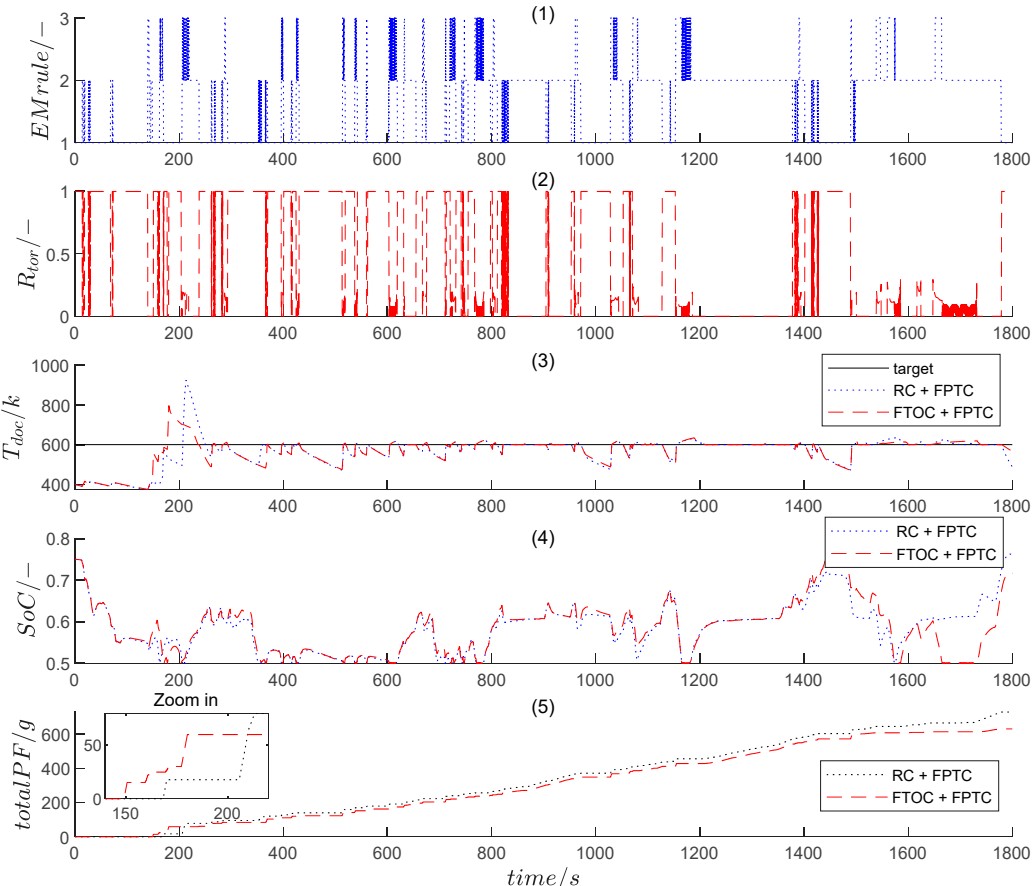

**Figure 14.** Comparison results of the RC + FPTC vs. the FTOC + FPTC: (**1**) energy mode rule of the RC, (**2**) $R_{tor}$ of the FTOC, (**3**) $T_{doc}$, (**4**) $SoC$, (**5**) total fuel consumption.

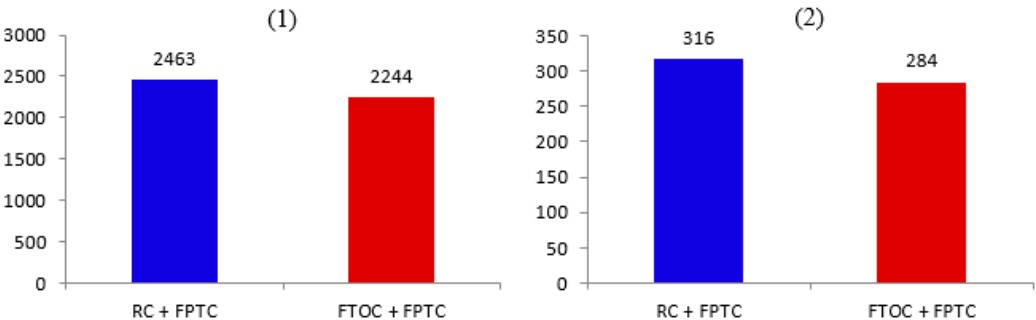

**Figure 15.** Comparison results of the RC + FPTC vs. the FTOC + FPTC: (**1**) total fuel consumption (g), (**2**) total $NO_x$ emission (g).

### 6. Evaluation for the Information of Road Grade on the DHEV

Road grade is common urban conditions, especially the increasing entrance and exit grades of urban viaducts. Road grade will seriously interfere with the cruising, fuel consumption and emissions control of the DHEV. In this section, information on a fuel and thermal sensor optimized controller (FTSOC) for power split which obtains real-time information on the road grade (assuming that the vehicle is equipped with a grade sensor) is proposed, and a fuel and thermal slope forecast optimized controller (FTSFOC) for power split which obtains the preview information is also proposed. As shown in Equation (7), the method utilizes real-time road grade $\alpha$ to make a linear correction with $k_a$ for the power split factor $R_{tor}$ solved by the optimization problem as Equation (6).

$$R_{tor} = R_{tor} + k_a * \alpha. \tag{7}$$

Road grade information $\alpha(i)$, $i = 1 \ldots Np$ can be obtained in advance by intelligent cruise systems. With regard to $\alpha(i)$, $i = 1 \ldots Np$, the optimization-based method can optimize the linear relationship coefficient between $R_{tor}$ and $\alpha$ by solving $u_1$ and $u_2$ as shown in Equation (8).

$$R_{tor}(i) = u_1(i) + u_2(i) * \alpha(i). \tag{8}$$

As shown in Figure 16, under the cruising state where the vehicle speed objective is 40 mile/h, a comparison between the three controllers on the road grade interference is conducted. At the 60–90 s and 120–150 s phases, the $R_{tor}$ value of the FTOC is the smallest, and its proportion of the engine load is the largest. The torque output response of the engine is slower than that of the motor. When encountering a sudden torque demand on uphill slopes, the speed tracking of the FTOC has large fluctuations. This will also lead to an increase in the total torque output for tracking cruising speed. Although the proportion of the motor load is small, the total power consumption of the motor of the FTOC has increased. The *SoC* of the FTOC fluctuates the most and the average fuel consumption during the cruising process is 21.54 mg/cycle. The FTSOC can obtain the road grade information in real time and actively increases the proportion of the motor load, so that the response of power output is improved, and the fluctuations of the speed tracking are reduced. The average fuel consumption of the FTSOC during the cruising process is 20.36 mg/cycle, which saves nearly 5.5% compared with the FTOC. The FTSFOC can obtain the road grade information in advance with intelligent cruise systems and optimize $R_{tor}$ in accordance with the road grade in the prediction horizon. The $R_{tor}$ of the FTSFOC is adjusted most frequently, and the speed tracking of the FTSFOC has the smallest fluctuations. The *SoC* variance of the FTSFOC is always the same as the FTSOC. The average fuel consumption of the FTSFOC during the cruising process is 19.30 mg/cycle, which further saves nearly 5.2% compared with the FTSOC.

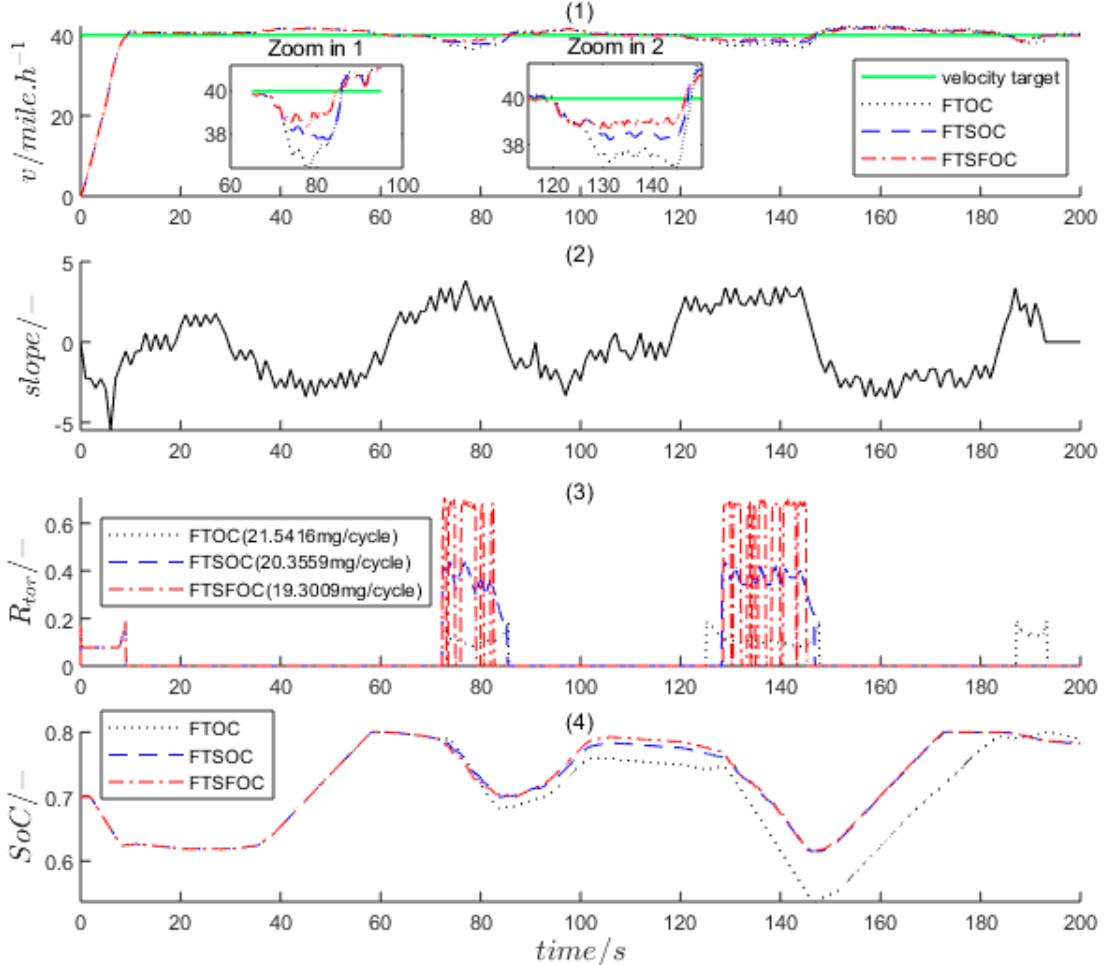

**Figure 16.** Comparison results of the FTOC vs. the FTSOC vs. the FTSFOC: (**1**) the speed target of the cycle and the tracking states of the controllers, (**2**) road grade, (**3**) $R_{tor}$, (**4**) *SoC*.

## 7. Conclusions

In this paper, a hierarchical model predictive control (MPC) framework is proposed to coordinate the power split and the exhaust thermal management. Preview information about road grade is also introduced to improve the power split by a fuel and thermal on slope forecast optimized controller (FTSFOC). Simulation results show that the hierarchical method (FTOC + FPTC) can reach the optimal exhaust temperature nearly 40 s earlier, and its total fuel consumption is also reduced by 8.9%, as compared to the sequential method under a world light test cycle (WLTC) driving cycle. Moreover, the total fuel consumption of the FTSFOC is reduced by 5.2%, as compared to the fuel and thermal on sensor-information optimized controller (FTSOC) working with a real-time road grade information. Our future studies will focus on using real-time calculation methods to solve the optimization problems, and performing experimental verification for the control strategy proposed in this paper.

**Author Contributions:** Conceptualization, J.Z. and X.G.; methodology, J.Z. and F.X.; software, X.L. and H.S.; validation, Y.H., Y.S. and X.G. All authors have read and agreed to the published version of the manuscript.

**Funding:** This research was funded by the National Nature Science Foundation of China (Grant Nos. 61773009, 61903152, U1864201, 62103160) and the Jilin Province Science and Technology Development Plan (20210203102SF) and the Foundation of State Key Laboratory of Automotive Simulation and Control (20191201) and the Science and Technology Project of Jilin Provincial Education Department (JJKH20210786KJ, JJKH20211098KJ, JJKH20200986KJ, 2018C034-5).

**Conflicts of Interest:** The authors declare no conflict of interest.

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
