# Peer review of "Modeling and Integrated Optimization of Power Split and Exhaust Thermal Management on Diesel Hybrid Electric Vehicles"

_energies, doi:10.3390/en14227505_

Round 1

Reviewer 1 Report

In this paper, the Authors compare different control strategies for hybrid powertrain optimization. The topic is relevant and the research is generally sound. However, the manuscript quality must be improved before publication. Please find below a list of major suggestions:

  1. Background should be improved through a deeper literature analysis. Many research branches in the field of optimized vehicles control strategy are not acknowledged. The Authors might start from the following papers to improve their literature:
    • Simona Onori, Laura Tribioli, Adaptive Pontryagin’s Minimum Principle supervisory controller design for the plug-in hybrid GM Chevrolet Volt, Applied Energy, Volume 147, 2015, Pages 224-234, ISSN 0306-2619, https://doi.org/10.1016/j.apenergy.2015.01.021.
    • Laura Tribioli, Michele Barbieri, Roberto Capata, Enrico Sciubba, Elio Jannelli, Gino Bella, A Real Time Energy Management Strategy for Plug-in Hybrid Electric Vehicles based on Optimal Control Theory, Energy Procedia, Volume 45, 2014, Pages 949-958,ISSN 1876-6102,
    • L. Serrao, S. Onori, A. Sciarretta, Y. Guezennec and G. Rizzoni, "Optimal energy management of hybrid electric vehicles including battery aging," Proceedings of the 2011 American Control Conference, 2011, pp. 2125-2130, doi: 10.1109/ACC.2011.5991576.
  2. In Eq. on Line 170, clarify the role of Pg. Is the generator draining power from the battery?
  3. Using single injection strategy is questionable when focusing on engine emissions. In fact, multiple injections greatly reduce the engine NOx and PM emissions, while moderately impacting its efficiency.
  4. What does the isolines in Figure 4 represtent? Please also specify units.
  5. Units of measure on the iso-lines of Figure 7 are missing
  6. Please include a paragraph describing the basic characteristics of the engine model.
  7. Equation (1): Why only radiative heat transfer and not convective heat transfer? Considering the typical engine exhaust temperature one would expect that convective transport is relevant. Please comment. 
  8. Please provide the source for Fig. 6 data.
  9. Using the dot for multiplication in the torque unit of measure in formally incorrect. It denotes scalar product between vectors.
  10. Please provide the values for all the model parameters (specifically for eq. 3)
  11. Figure 1 and 9 can be merged.
  12. In general the model description is unclear, and a little confusing. I suggest revising the paper avoiding general sentences such as " Due to the fuel consumption sensitivity to the separate fuel injection system, the coordinated exhaust thermal management between engine exhaust and the separate fuel injection should be careful" (line 301-302) while focusing on simple, linear and clear descriptions of the models and algorithms.
  13. How are the weights in the first of equations (4) determined.
  14. Please improve quality of figure 12
  15. It is unclear whether the hybrid control strategy can be implemented in real time on vehicles.
  16. Please improve conclusions including some hints on possible future development and on the applicability of the proposed control strategies for real applications. 
  17. Language could be improved, in particular in terms of syntax and text organization.

Author Response

First we would like to express sincere thanks to the Assistant Editor and anonymous reviewers for their time and efforts, and constructive comments and suggestions. We would be happy to resubmit our substantially revised manuscript for further assessment. We have carefully considered those comments and suggestions, and thoroughly revised and improved the manuscript with added analysis and evaluation. A detailed description is given in the following. In the manuscript the comments for reviewer 1 are all highlighted in blue. Please see the attachment.

Reviewer 2 Report

The manuscript is interesting and considers a topic of interest for the journal. It should be considered after establishing a series of changes that affect the content and organization of the work. 
The manuscript should be revised to try to eliminate the coincidences established after passing through anti-plagiarism programs (TURNITIN) that show a percentage of similarity of 21% (this file is attached). 
The length of the manuscript should be reduced, considering that the introduction considers aspects irrelevant to the study, which could be summarized and/or deleted. Another section that should be summarized is section 2, considering only general aspects of each part. The work should focus more on the modeling of the established method, without so many sub-sections that make it difficult to understand the process followed.
The comparison and discussion of the established results should be clearer, as well as the conclusions that are scarce and incomplete.
The quality of the figures should be improved (Figures 5, 10, 12, 15, 17), considering the lack of resolution of the legends, in addition to being properly edited, it should be considered that the colors chosen should be more in accordance with the journal, avoiding dark colors such as red or orange that make them difficult to understand (Figures 1, 2, 3, 9, 13, 14).
All the mathematical expressions must be properly referenced in the text, not all the expressions established before and after the marked equations have been considered. 

Author Response

First we would like to express sincere thanks to the Assistant Editor and anonymous reviewers for their time and efforts, and constructive comments and suggestions. We would be happy to resubmit our substantially revised manuscript for further assessment. We have carefully considered those comments and suggestions, and thoroughly revised and improved the manuscript with added analysis and evaluation. A detailed description is given in the following. In the manuscript the comments for reviewer 2 are highlighted in green.

Reviewer 3 Report

Manuscript review: Modeling and Integrated Optimization of Power Split and Exhaust Thermal management on Diesel Hybrid Electric Vehicles  (energies-1402053).

The manuscript addresses the hierarchical model predictive control (MPC) frame work is proposed to coordinate the power split and the exhaust thermal management.

  1. The abstract is well written. I have no reservations. Just add the issue of verification of tests - simulations only or tests on a chassis dynamometer?
  2. A review of literature is absolutely not enough. Please highlight the knowledge gap, justify the purpose of your research, and write how it differs from other authors? The literature presented in the manuscript does not include the latest publication achievements. There are a large number of manuscripts (out in Energies) in which the authors examine vehicles equipped with different engines and running on different fuels in the context of driving tests (computer simulations and verifications on chassis dynamometer).
  3. I would very much appreciate the addition of a table listing all the symbols used in the manuscript.
  4. The simulation is well done and I have no reservations. However, I would suggest referring to a specific driving test and driving requirements (WLTC driving cycle - please describe it better). What about allowable emission limits?
  5. Did the authors consider LCA analysis?
  6. The conclusions of the studies carried out need to be supplemented. They are very general and any discussion of them is lacking.

Author Response

First we would like to express sincere thanks to the Assistant Editor and anonymous reviewers for their time and efforts, and constructive comments and suggestions. We would be happy to resubmit our substantially revised manuscript for further assessment. We have carefully considered those comments and suggestions, and thoroughly revised and improved the manuscript with added analysis and evaluation. A detailed description is given in the following. In the manuscript the comments for reviewer 3 are highlighted in red.

Reviewer 4 Report

A potentially very interesting and informative topic. My greatest concerns relate to the Introduction and to the study's increment.

I personally find the Introduction to be lacking in logical flow through to the background foundation and arguments that it is attempting to develop. I had to look to the Abstract and the Conclusion section to identify the objective(s) of the study. The Introduction needs to be much better structured and clearer.

The study needs to explain/argue why from an ex ante persepective, considering the two approaches in combination might have the possibility of leading to an improvement (over each).

At presented, I don't believe that the Introduction clearly lays out and confirms the incremental contribution of the study given the extant literature (especially that cited in the Introduction). This aspect of the study needs to be strengthened and made much more explicit. Is there in fact a true tension, or are the findings in many ways obvious (such as 'road grade').

Given statements in the Introduction around current approaches, I believe that it is incumbent on the authors to also present a discussion around translation of their findings from the 'laboratory simulation' environment to 'real world application and performance'.

Again, a very interesting and potentially informative choice of topic.

Author Response

First we would like to express sincere thanks to the Assistant Editor and anonymous reviewers for their time and efforts, and constructive comments and suggestions. We would be happy to resubmit our substantially revised manuscript for further assessment. We have carefully considered those comments and suggestions, and thoroughly revised and improved the manuscript with added analysis and evaluation. A detailed description is given in the following. In the manuscript the comments for reviewer 4 are highlighted in orange.

Round 2

Reviewer 3 Report

The current version of the manuscript is very good.

I do not have any remarks.

I recommend it for publication.